Incomplete lineage sorting and ancient admixture, and speciation without morphological change in ghost-worm cryptic species

Cerca José jose.cerca@ntnu.no 1 2 3
Rivera-Colón Angel G. 4
Ferreira Mafalda S. 5 6 7
Ravinet Mark 8 9
Nowak Michael D. 3
Catchen Julian M. 4
Struck Torsten H. 3
1 Department of Environmental Science, Policy, and Management, University of California, University of California, Berkeley , Berkeley , CA , United States of America
2 Department of Natural History, NTNU University Museum, Norwegian University of Science and Technology , Trondheim , Norway
3 Natural History Museum, University of Oslo , Oslo , Norway
4 Department of Evolution, Ecology, and Behavior, University of Illinois at Urbana-Champaign , Urbana Champaign , IL , United States of America
5 Division of Biological Sciences, University of Montana , Missoula , MT , United States of America
6 Departamento de Biologia, Universidade do Porto , Porto , Porto , Portugal
7 CIBIO, Centro de Investigação em Biodiversidade e Recursos Genéticos, InBIO Laboratório Associado, Universidade do Porto , Porto , Porto , Portugal
8 School of Life Sciences, University of Nottingham , Nottingham , United Kingdom
9 Centre for Ecological and Evolutionary Synthesis, University of Oslo , Oslo , Norway
Gillespie Joseph
Electronic publication date: 2021 Feb 9
Publication date: 2021
Volume: 9
Electronic Location ID: e10896
Received 2020 Sep 15; Accepted 2021 Jan 13
Copyright: ©2021 Cerca et al.
Copyright year: 2021
Copyright holder: Cerca et al.
License: This is an open access article distributed under the terms of the Creative Commons Attribution License, which permits unrestricted use, distribution, reproduction and adaptation in any medium and for any purpose provided that it is properly attributed. For attribution, the original author(s), title, publication source (PeerJ) and either DOI or URL of the article must be cited.
License URL: https://creativecommons.org/licenses/by/4.0/

Keywords: Morphology, Rapid evolution, Slow evolution, Stasis, Genomics, RADseq, Phylogenetics, Morphological evolution, Genomics, Sibling species

Funding: Den Grevelige Hjelmstjerne-Rosencroneske Stiftelse ved UiO UiO:LifeScience internationalization support Godfrey Hewitt mobility ESF Portuguese MCTES/FCT NSF OIA-1736249 Fieldwork was supported by internal funds, a Den Grevelige Hjelmstjerne-Rosencroneske Stiftelse ved UiO (José Cerca), and THS was supported by the EU Assemble program. A UiO:LifeScience internationalization support and a Godfrey Hewitt mobility award allowed José Cerca to visit Julian Catchen at the UIUC. MSF was supported by POPH-QREN funds from ESF and Portuguese MCTES/FCT (PD/BD/108131/2015 PhD grant in the scope of BIODIV PhD programme at Faculty of Sciences, University of Porto) and NSF (OIA-1736249). There was no additional external funding received for this study. The funders had no role in study design, data collection and analysis, decision to publish, or preparation of the manuscript.

==============================
Morphologically similar species, that is cryptic species, may be similar or quasi-similar owing to the deceleration of morphological evolution and stasis. While the factors underlying the deceleration of morphological evolution or stasis in cryptic species remain unknown, decades of research in the field of paleontology on punctuated equilibrium have originated clear hypotheses. Species are expected to remain morphologically identical in scenarios of shared genetic variation, such as hybridization and incomplete lineage sorting, or in scenarios where bottlenecks reduce genetic variation and constrain the evolution of morphology. Here, focusing on three morphologically similar Stygocapitella species, we employ a whole-genome amplification method (WGA) coupled with double-digestion restriction-site associated DNA sequencing (ddRAD) to reconstruct the evolutionary history of the species complex. We explore population structure, use population-level statistics to determine the degree of connectivity between populations and species, and determine the most likely demographic scenarios which generally reject for recent hybridization. We find that the combination of WGA and ddRAD allowed us to obtain genomic-level data from microscopic eukaryotes (∼1 millimetre) opening up opportunities for those working with population genomics and phylogenomics in such taxa. The three species share genetic variance, likely from incomplete lineage sorting and ancient admixture. We speculate that the degree of shared variation might underlie morphological similarity in the Atlantic species complex.

Introduction

The characterization and delimitation of species and populations using DNA sequencing and barcoding has led to the discovery of ‘hidden species diversity’ in previously established species (Knowlton, 1993; Bickford et al., 2007; Pfenninger & Schwenk, 2007; Struck et al., 2018). The initial interest in this hidden diversity, that is cryptic species, fuelled a debate on whether these lineages resulted from biases of a morphologically oriented classification of biodiversity or whether they resulted from underlying biological phenomena. On one side, proponents of the “artefact model” suggest that populations and species naturally accumulate morphological differences, and it is only the limitations associated with scientific methods that impede the discovery of those differences (Korshunova et al., 2017). On the one other side, the “evolutionary framework” suggests that the deceleration of morphological evolution is a plausible expectation, given the observation of stasis, niche conservatism and constraints in nature. While some of this diversity is potentially attributed to taxonomic artefacts (Korshunova et al., 2017), morphologically similar species—‘true’ cryptic species—have been discovered in various branches of the tree of life, thus representing an important part of biodiversity (Pfenninger & Schwenk, 2007; Pérez-Ponce de León & Poulin, 2016; Cerca, Purschke & Struck, 2018; Fišer, Robinson & Malard, 2018).

Following centuries of morphologically oriented taxonomy, the existence of “true” cryptic species entails a challenge to the delimitation, discovery and classification of species (Bickford et al., 2007; Fišer, Robinson & Malard, 2018; Struck et al., 2018). In the case of morphologically similar species, species delimitation relying on morphology alone will fail to capture the existing species diversity (Pante et al., 2015; Fišer, Robinson & Malard, 2018; Chenuil et al., 2019; Struck & Cerca, 2019), resulting in the lumping of different species into a single species complex. While much has been written on the consequences of cryptic species in terms of biological systematics, we have only recently begun to understand the impact of cryptic species in other fields of biology. When species are poorly delimited, determination of biogeographic breaks (Weber, Stöhr & Chenuil, 2019; Cerca et al., 2020a), inferences on the evolutionary history (Wada, Kameda & Chiba, 2013; Swift, Daglio & Dawson, 2016; Struck et al., 2018; Dufresnes et al., 2019), and the determination of ecological richness of an ecosystem (Chenuil et al., 2019) may be severely compromised. These problems extend outside fundamental fields of biology when species complexes are medically-relevant, such as the Anopheles cryptic species complex where not every morphologically-similar species is capable of transmitting malaria (Erlank, Koekemoer & Coetzee, 2018) or in parasite species (De León & Nadler, 2010; Nadler & De Len, 2011), but also in cases of conservation management (Bickford et al., 2007; Bernardo, 2011).

While the discovery of cryptic species complexes has increased in the last few years, the resulting debate has focused on whether these are taxonomic artefacts or biologically relevant species. Consequently, the causes underlying morphological similarity remain mostly unexplored. Despite this hindrance, an important source of information may come from palaeontology where stasis has been studied for decades (Eldredge & Gould, 1972; Gould, 2002), and from the subsequent integration of this evidence with neontological data. A particularly insightful contribution is that of Futuyma (2010), which suggests that stasis may result from certain ecological, genetic and developmental scenarios. Genetic scenarios include shared genetic variation, potentially resulting from hybridization or ILS, homogenizing morphological divergence; genetic constraints resulting from epistatic reactions or pleiotropy, or constrains from the lack of genetic variation due to repeated bottlenecks or founder effects; stabilizing selection on morphology (Futuyma, 2010). Some of these scenarios including stabilizing selection (Lee & Frost, 2002; Novo et al., 2010; Novo et al., 2012; Lavoué et al., 2011; Smith et al., 2011; Santamaria et al., 2016; Zuccarello, West & Kamiya, 2018), bottlenecks and founder effects (Dornburg et al., 2016; Valtueña et al., 2016) have been proposed to explain similarity on cryptic species. However, this remains untested since evidence for morphological similarity comes mostly from the interpretation of indirect methods, such as phylogenetic trees.

The Stygocapitella genus includes 11 described species with only four morphotypes and no significant quantitative morphological differences between some species (Cerca et al., 2020a; Cerca et al., 2020b). Morphologically identical species occur in sympatry and overlap in their distribution along the Northern European, Atlantic American, and Pacific American coastlines. In a previous study, we confirmed that three North Atlantic species—Stygocapitella westheidei, S. subterranea, and S. josemariobrancoi—are morphologically identical (Cerca et al., 2020a); nonetheless, we were not able to determine the causes underlying morphological similarity with certainty. Preliminary results from selected DNA markers indicated that morphological similarity potentially stems from niche conservatism and niche tracking, coupled with the fluctuating dynamics of their habitats and/or genetic constraints (Cerca et al., 2020b). Here, using genomic data, we extend these efforts by focusing on the causes linked to genetic variation underlying morphological similarity (see above). Following Futuyma (2010), we hypothesize that (1) bottlenecks and founder effects reduce genetic variation, thus resulting in morphological similarity; (2) morphological similarity results from recent admixture; (3) shared genetic variation due to incomplete lineage sorting and ancient admixture underlies morphological similarity.

Methods and Materials

Study system

Stygocapitella is part of the meiofauna, being generally found above the high-water line of sheltered gravel or sandy beaches. To collect individuals, we selected sampling areas based on old records or by assessing beaches using google maps (Table S1; Fig. 1). At each site, we drew a transect from the high-water line to the foot of the dune, digging a 1-meter deep hole every meter starting at the high-water line. In each hole, we collected sediment samples every 15 cm of depth with a volume of about 500 cm3. Sediment samples were brought to the laboratory and interstitial invertebrates were extracted using the MgCl2 method, and isolated using a dissecting microscope (Westheide & Purschke, 1988). After identifying Stygocapitella, we collected and preserved these in a ∼70% ethanol solution for DNA extraction.

Figure 1 Sampling locations across the Northern Atlantic.

(A–B) North America, (C) United Kingdom, France and Germany, and the Island of Sylt in Germany. The three species are displayed in different colours: orange (Stygocapitella westheidei), green (S. josemariobrancoi) and blue (S. subterranea). Sampling locations with multiple circles denote populations in sympatry. Species are delimited using the COI, 16S, 18S and ITS1 barcodes.

DNA extraction and molecular species barcoding

Since Stygocapitella westheidei, S. subterranea and S. josemariobrancoi are morphologically indistinguishable (Cerca et al., 2020a), we barcoded individual individuals using 16S, 18S, ITS1 and COI as described in Cerca et al. (2020b) and Cerca et al. (2020a) (Table S2 includes NCBI reference-IDs). In brief, we extracted DNA from single individuals using either phenol-chloroform or the E.Z.N.A. Tissue DNA Kit (Omega Bio-Tek), and obtained COI (Astrin & Stüben, 2008), 18S (Hillis & Dixon, 1991), ITS1 (Cerca et al., 2020a), and 16S (Palumbi et al., 1991; Zanol et al., 2010) sequences using PCR. Amplified genetic markers were sequenced by Sanger-sequencing at Macrogen-Europe. For detailed information on amplification, primer sequences and extraction please see Cerca et al. (2020b) and Cerca et al. (2020a).

Library preparation and Illumina sequencing

We selected 50 Stygocapitella josemariobrancoi, 47 S. subterranea and 24 S. westheidei for library preparation (Table S2). Due to the reduced body size, DNA extractions of Stygocapitella yield low concentrations of DNA, therefore, to overcome this problem, we used a combination of whole genome amplification (WGA) (Golombek et al., 2013; De Medeiros & Farrell, 2018) followed by a double-digestion Restriction site-Associated DNA sequencing protocol (ddRAD) (Baird et al., 2008; Peterson et al., 2012). To complete the WGA reaction, DNA of a single individual is first denaturated and mixed with random hexamer primers and the Phi29 DNA polymerase (Illustra Genomiphi HY DNA Amplification Kit; GE©Healthcare Life Science). Following the manufacturer’s instructions, 2.5 µl of template DNA were mixed with 22.5 µl of sample buffer, and incubated at 95 °C for three minutes for denaturation. After this, we added 22.5 µl of reaction buffer and 2.5 µl of enzyme mix to the DNA-sample buffer solution, incubated the solution at 30°C for four hours for DNA amplification and an enzyme heat-inactivation at 65 °C for ten minutes. DNA was purified using AMPure XP beads, and resuspended in ddH2O. The concentration of the amplified DNA was determined with Qubit and the fragment size distribution with a fragment-analyzer.

For each individual, 500 ng of amplified DNA was digested in 25 µl including 0.5 µl of each restriction enzyme (Pst-I HF and Mse-I, each 20 units/µl) and 2.5 µl Cut-smart buffer. The digestion reaction was carried out at 37 °C for two hours. Digested DNA was purified using Ampure-beads and resuspended in 22 µl ddH2O, and Illumina adaptors with barcodes were ligated to the digested DNA in a 25 µl reaction including 20.5 µl sample DNA, 1 µl T4 DNA-ligase, 2.5 µl 10X T4 ligase buffer and 1 µl adapter P1/2-mix. This reaction was incubated for 30 min at 25 °C, and the enzyme inactivated for 10 min at 65 °C. The barcoded libraries were pooled, cleaned using AMPure XP beads, and eluted in 100 µl of ultra-purified water. We ran a size-selection step using Blue Pippin’s 100–600 bp cassette (BDF2010) selecting for fragment-length between 300–600 bp followed by cleaning with AMPure XP beads to remove short fragments. The library was amplified in 200 µl including 100 µl Q5 HiFi MasterMix, 5 µl Primer mix and 20 µl DNA, in 18 PCR cycles (initial denaturation: 98 °C for 30s; 18 cycles of 98 °C for 10 s, 60 °C for 15 s, 72 °C for 15 s; and a final elongation of 72 °C for 2 min). Finally, two cleaning-quantification steps using AMPure XP beads were done, and the libraries were sent for Illumina Sequencing on an Illumina Hi-Seq 4000.

Individual identification and multi-marker phylogeny

Raw 16S, 18S, ITS1 and COI sequences were assembled, and ends were automatically trimmed to remove primers and low-quality ends using Geneious v6.8.1, (Table S2). Each consensus sequence was queried against the default NCBI database (nr/nt) using BLAST (Altschul et al., 1990; Camacho et al., 2009) to exclude potential contamination. For each gene, sequences were aligned using mafft v7.310, using a maximum of 1,000 iterations, and the ends of the sequences trimmed until the first position without missing data. The accurate localpair algorithm was used for all genes (Katoh & Standley, 2013), with the exception of ITS1, which had a single peak, where the globalpair algorithm was applied as it is optimized for gappy sequences. The dataset was concatenated using FASconCAT v1.1 (Kück & Meusemann, 2010), and a partitioned phylogenetic tree was obtained using IQ-tree v1.6.10, by applying 1,000 fastbootstrap replications. Model determination was done automatically by IQ-tree, and it included TIM2+I for ITS1, JC+I for 18S, TN+R2 for 16S and HKY+I for COI. The congruence between these genes, and in the individuals used has previously been determined in Cerca et al. (2020b) and Cerca et al. (2020a).

De novo RAD assembly

All bioinformatic work is available in https://github.com/jcerca/Papers/tree/main/Stygocapitella_PeerJ. Since no reference genome is available for Stygocapitella, we used the de novo assembly approach implemented in Stacks v2.2 to identify RAD loci (Rochette & Catchen, 2017; Rochette, Rivera-Colón & Catchen, 2019). The first module of Stacks, ‘process_radtags’, was executed using flags tailored to improve data quality (-clean, -quality, -rescue). To optimize Stacks’ parameters, we ran the de novo pipeline repeatedly using different -M (mismatch between stacks within individuals) and -n (mismatches between stacks between individuals) values, as suggested by best practices (Paris, Stevens & Catchen, 2017). The total number of loci resulting from different -M -n values were plotted and analyzed, selecting -M 3 and -n 3 for the final dataset. Populations, as required for the population map, were defined based on the species and sampling site (total of 22 populations—Table S1).

Since we observed a considerably high level of missing data in the dataset (>90% missing data), we tested and implemented a new method to improve RADseq datasets. Missing data is especially problematic in RADseq as it can lead to erroneous inference of population-genetic parameters (Arnold et al., 2013; Gautier et al., 2013; Hodel et al., 2017). However, applying stringent filtering for missing data has been shown to prune parsimonious-informative loci, with best-practices suggesting non-conservative pruning of the data (Huang & Lacey Knowles, 2016; Lee et al., 2018; Crotti et al., 2019). To mitigate missing data while avoiding stringent filtering, we applied a novel procedure, which allowed us to retrieve more loci from our data (Cerca et al., 2021). In brief, we ran Stacks for every population present in the population map—22 times in total (Table S1)—thus lowering phylogenetic distance in the dataset. Since phylogenetic distance (biological origin) and artefacts in generating and processing data can lead to allelic dropout (O’Leary et al., 2018), lowering phylogenetic distance will isolate dropout caused by artefacts in library preparation (e.g., DNA size-selection, low DNA concentrations, poor digestion), and loss of information due to whole genome amplification (De Medeiros & Farrell, 2018). For each population, we identified samples with >45% missing loci and removed these from a final analysis (hereafter the clean dataset). To evaluate how the optimization impacted the final number of loci we compared the number of loci and missingness in the dataset before cleaning (hereafter the uncleaned dataset) and in the cleaned dataset using the values of -r 25 (a locus has to be present in at least 25% of individuals comprising a population to be considered) and -p 4 (a locus has to be present in at least 4 populations to be considered). Finally, we included a technical replicate in the dataset (individuals 222 01 and 222 01R), and checked whether it was coherently placed in all the analyses.

Population genomics and phylogenomics

From the clean dataset, we extracted a single nucleotide polymorphism-only dataset (SNPs; hereafter variant dataset) and an all-sites dataset (containing non-variant and variant positions; hereafter all-sites dataset). Separating the data in these two datasets is necessary to meet the assumptions of some statistical tests which may require the presence of non-variant positions to calculate ratios of variant and non-variant sites. The variant dataset was pruned by selecting -r 50 (a locus has to be present in at least 50% of individuals comprising a population), and -p 8 (a locus has to be part in at least 8 populations) as loci cut-offs, using the ‘populations’ program included in Stacks, resulting in 4,737 RAD-loci. After this initial round of cleaning, we used vcftools v0.1.13 (Danecek et al., 2011) to further prune the dataset for 5% minimum allele frequency (–maf), and for mean loci coverage values between 10–100 (–min-meanDP 10 –max-meanDP 100) and removed 12 individuals which had missingness above >90% (–missing-indv; Table S2). The combination of coverage filters together with the -M -m optimization procedure mentioned above optimizes the generation of RADseq loci by removing loci which may artificially come together (i.e., repetitive regions). Finally, to decrease the effect of physical linkage in the data, we used a custom BASH script which kept only one polymorphism (SNP) per RAD locus, resulting in a final dataset of 3,428 SNPs. Using this dataset, we assessed genetic variation by means of a principal component analysis (PCA), a multi-dimensional scaling (MDS) analysis and an ADMIXTURE analysis. PCA and MDS are model-free approaches to estimate population structure, being complementary as PCA assumes ‘mean values’ for missing data (i.e., dragging individuals with high missingness to the center) whereas MDS does not. PCA was computed using the R package Adegenet (Jombart & Ahmed, 2011) and MDS with plink v1.9 (Chang et al., 2015). ADMIXTURE is a model-oriented approach to determine population structure based on the presence/absence of heterozygotes (Alexander, Novembre & Lange, 2009). We ran ADMIXTURE assuming 1-6 clusters (K), running a total of 5 replicates for each K, and determined the best K by estimating the cross-validation error (Fig. S1). Considering the potential for admixture of individuals in sympatry (Fig. 1), we used f3 statistics, included as part of the Admixtools package, as a direct test for detecting hybridization (Patterson et al., 2012; Peter, 2016). These statistics consists of a 3 populations test where a focal population is derived from admixture between the other two populations. When this score is negative, it suggests that admixture likely has occurred. We estimated errors and confidence intervals on the f3 statistics by partitioning the dataset into blocks and applying a jackknife bootstrapping. Finally, we inferred species-level divergence by estimating Weir and Cockerham’s FST using vcftools (Weir & Cockerham, 1984; Danecek et al., 2011).

The all-sites dataset was obtained by extracting FASTA sequences from Stacks. To run the phylogenomic analysis, we wrote a Perl script to reorganize the data into loci, concatenated all loci in a supermatrix using FasConCat-G (Kück & Longo, 2014) and ran a partitioned-tree using IQ-tree v1.6.10 (Nguyen et al., 2015) specifying 1,000 fast bootstrap replications (Chernomor, Von Haeseler & Minh, 2016; Hoang et al., 2017), and locus-specific models which were determined as part of the run (Kalyaanamoorthy et al., 2017). To explore the effect of missing data on the tree topology, we ran BaCoCa (Kück & Struck, 2014), a pipeline which provides summary-level statistics on the concatenation matrix and tree, such as the % of positions with missing data shared by a pair of individuals. Pairwise positive overlap values were plotted to the tree topology using the R package ape (Paradis & Schliep, 2018). Additionally, we ran an Unweighted Pair Group Method with Arithmetic mean (UPGMA) tree using only the average % of pairwise shared data per individual (i.e., pairwise percentage of shared data between taxa which do not have an indel, ambiguous character state, or a missing character state). The UPGMA tree was run to understand whether taxa were grouped based on the overall pattern of missing data. Additionally, since RADseq loci (represented by allele 0 and allele 1) are not phased and since the labelling of 0 and 1 are arbitrary, we obtained a consensus sequence for each individual. This was done by running the consambig (consensus ambiguous) module included in the EMBOSS pipeline (Rice, Longden & Bleasby, 2000). We did a species network analysis using SPLITSTREE v4 (Huson & Bryant, 2006) to complement phylogenetic inference since the network does not enforce dichotomous branching at each node.

To gauge population-level patterns and diversity, we selected loci from the all-sites dataset without missing-data at the population-level and estimated summary statistics including nucleotide diversity (π), Waterson’s estimator of genetic diversity (S) and Tajima’s D using DNAsp v6 (Rozas et al., 2017). This dataset was not pruned for minimum allele frequency. The selection of sites without missing data is grounded on best-practices as missing data can lead to the under- or overestimation of some of these parameters (Arnold et al., 2013). Importantly, we selected only populations where 3 or more individuals (i.e., >5 ‘loci’) had data available (Table S2).

Finally, we evaluated various demographic scenarios using fastsimcoal2, using the same dataset for the previous analysis which included running fastsimcoal2 (Excoffier et al., 2013). Fastsimcoal2 uses the site-frequency spectrum (SFS) and a coalescent-simulation framework based on an arbitrary user-defined scenario to infer population sizes, strength of gene flow and times of coalescence. To assess these models we calculated AIC and likelihood. Likelihood is calculated by running the ‘best parameters’ for each specified scenario multiple times and obtaining the distribution of likelihood estimates. AIC was calculated using a script available in https://speciationgenomics.github.io/fastsimcoal2/. To implement these simulations, we used the phylogeny obtained with Stygocapitella subterranea and S. westheidei as sister species, and S. josemariobrancoi as sister to the remaining two. We defined the following models: no gene flow, ancient gene flow (i.e., between Stygocapitella josemariobrancoi and the stem lineage of S. subterranea and S. westheidei), geographic gene flow (i.e., similar as the ancient gene flow, but also with modern gene flow between the sympatric Stygocapitella subterranea and S. josemariobrancoi; note we refer to ‘modern’ as opposed to ‘ancient’, that is, in existing lineages), modern gene flow (gene flow between all three modern species), all gene flow (gene flow between all three species and the two ancestral lineages), modern gene flow only between S. josemariobrancoi and S. subterranea, modern gene flow only between S. josemariobrancoi and S. westheidei, modern gene flow only between S. westheidei and S. subterranea, gene flow in ancestral times and between S. subterranea and S. westheidei, and ancient gene flow and modern gene flow between S. josemariobrancoi and S. westheidei. When included in the model, gene flow was moduled as asymmetric. Each model was run 10,000 times with an assumed mutation rate of 1.2e−8, and the best fitting scenario was evaluated using likelihood, by running it 100 times.

Figure 2 Phylogenetic reconstruction and Scanning Electron Microscopy images of Stygocapitella.

(A) Maximum likelihood phylogeny of a concatenated, partitioned dataset (COI, 16S, 18S and ITS1), with scale provided on the bottom. Coloration follows species, with blue representing Stygocapitella subterranea, green representing S. josemariobrancoi, and orange S. westheidei. Bootstrap support for the branches representing species are provided on top of the branches. Every species is retrieved as monophyletic. (B) Drawing of the Stygocapitella westheidei, S. subterranea, S. josemariobrancoi morphotype. For more information on the classification and distinction of the morphotypes see Cerca et al. (2020a) and Cerca et al. (2020b). (C) Phylogenomic tree based on 4,737 RADseq loci. Bootstrap support is provided for the main branches.Coloration follows species with blue representing Stygocapitella subterranea, green representing S. josemariobrancoi, and orange S. westheidei. Three specimens, denoted by arrows, are identified as ‘paraphyletic’, including Bristol Channel 422 04 (identified as S. josemariobrancoi, nested within S. subterranea), Bristol Channel 422 05 (identified as S. josemariobrancoi, nested within S. westheidei) and St. Efflam 401 03 (identified as S. josemariobrancoi, nested sister to S. subterranea). The tree topology is coloured with shared pairwise data as estimated by BaCoCa. Allele 0 and allele 1 are displayed for all specimens. Specimen 222_01 is a technical replicate and is therefore represented twice. Shared pairwise data was calculated by integrating BaCoCa’s information on pairwise missing data.

Results

Tree of selected molecular markers

We compiled a dataset comprising 4,147 bp (the COI fragment consisted of 629 bp, 16S of 548 bp, ITS1 of 1,150 and 18S of 1,817 bp), from which 716 sites were phylogenetic-informative sites. From a total of 69 individuals, we obtained 67 16S sequences, 61 COI sequences, 28 18S sequences and 31 ITS1 sequences. Every species was recovered as monophyletic (Fig. 2A) with bootstrap support values of 100 for S. josemariobrancoi, of 86 for S. subterranea and of 100 for S. westheidei (Fig. 2A). The retrieved tree topology includes S. westheidei and S. subterranea as sister species. Stygocapitella josemariobrancoi as sister to the clade comprising S. subterranea and S. westheidei. Single gene trees show concordance between markers (Figs. S2–S5)

Genomic dataset

We obtained a total of 1,277,919,764 sequencing reads from two Illumina HiSeq4000 lanes. After demultiplexing and cleaning the data with process_radtags, we retained 899,112,800 reads (107,830,588 reads were discarded for having ambiguous barcodes, 270,174,154 for ambitious RADtags, and 802,222 for low quality). When comparing the clean and unclean dataset, the approach to lower allelic dropout yielded a substantial increase of the number of loci. In detail, after running Stacks for each population individually, we removed 17 S. subterranea out of a total of 47, 7 S. westheidei out of 24, and 16 out of 50 for S. josemariobrancoi (roughly ∼33% of the dataset, Table S2). The uncleaned dataset yielded 179,742 loci (55,037,190 sites including 628,031 variant sites), whereas the cleaned data yielded 368,696 loci (112,725,106 sites including 1,100,431 variant sites). When pruned with common denominators included in the module Populations (-r 50 -p4), the unclean dataset yielded 109,369 loci, and the cleaned yielded 272,134 loci. Individual-level missing data was reduced from 84.89% in the uncleaned dataset to 80.79% in the cleaned dataset. We validated this approach by comparing PCA, MDS and phylogenomic trees using both cleaned and the uncleaned dataset (see Cerca et al. (2021)). A comprehensive investigation of this strategy including additional datasets will be published in a separate paper (Cerca et al., 2021).

Genomic trees and networks

The phylogenomic tree (Fig. 2C) shows a slightly different topology from the tree obtained with selected molecular markers. The branches representing Stygocapitella subterranea, S. westheidei, and S. josemariobrancoi have a bootstrap support of 93, 99 and 100, respectively. The tree topology is broadly similar to the selected marker phylogenetic tree, with S. josemariobrancoi being sister to the clade comprising S. subterranea and S. westheidei. However, strictly speaking, none of the species is recovered as monophyletic, since three individuals identified as S. josemariobrancoi are not placed with S. josemariobrancoi. Specifically, 422 04 from Bristol Channel nests within S. subterranea, 422 05 from Bristol Channel nests within S. westheidei and 401 03 from St. Efflam is positioned as sister to S. subterranea (Fig. 2C, individuals denoted by arrows). Importantly, mapping of shared pairwise data in the tree topology suggests that these trends are not driven by missing data, since the branches representing the three aforementioned individuals do not exhibit elevated levels of missing data (Fig. 2C). The UPGMA tree, which is solely built on a pairwise matrix of missing-data, shows that S. subterranea and S. josemariobrancoi are generally separated, interlaced by individuals from S. westheidei (Fig. S6). While this suggests that the three species have different amounts of missing data, the fact that individuals are generally mixed suggests that missing data is not driving phylogenetic reconstruction. For example, the three individuals resulting in a paraphyletic reconstruction are not placed closely to their sister taxa in the tree, therefore indicating that missing data has no influence in the paraphyletic position of these individuals (Fig. S6). Finally, the generated phylogenomic consensus tree shows a similar topology to the that in Fig. 2 (Fig. S7). The three samples causing paraphyly of the lineages in the phylogenomic tree are placed within S. subterranea (Bristol Channel 422 04), as sister to the lineage S. josemariobrancoi and S. westheidei (St. Efflam 401 03), and as the first branch of S. josemariobrancoi (Bristol Channel 422 05).

In the phylogenetic network, Stygocapitella westheidei is separated from the remaining two species, occupying a separate and relatively compact area of the network. Stygocapitella subterranea is mostly confined to one small section of the network, however, three individuals are very close to the center of the network (398 04, 398 08 and 398 09 from Keitum; Fig. 3). In line with the results from the phylogenomic tree, Bristol Channel 422 04, which is identified as part of S. josemariobrancoi, is nested within S. subterranea in the network. Stygocapitella josemariobrancoi, on the other hand, is clearly stretched and set apart in the network, occupying a large area (Fig. 3). While most individuals are nested within a condensed and remote portion of the network, the individuals Bristol Channel 422 05, St Efflam 401 03 and 401 04 lie in an intermediate position between the center of the network and the majority of individuals from S. josemariobrancoi (Fig. 3). This is broadly in agreement with the phylogenomic tree, which shows Bristol Channel 422 05 nested with S. westheidei (Fig. 2C) and St. Efflam 401 03 sister to S. subterranea (Fig. 2C). The distance between most individuals belonging to S. josemariobrancoi and the rest of the network (centre of the network, and the two remaining species) suggests there is a greater degree of differentiation in this species.

Figure 3 Phylogenetic network based on 4,737 RADseq loci.

Coloration follows species with blue representing Stygocapitella subterranea, green representing S. josemariobrancoi, and orange representing S. westheidei. Specimens with arrows represent specimens which were as paraphyletic in the phylogenomic tree (Fig. 2C). S. josemariobrancoi is clearly stretched, indicating a greater differentiation from the remaining two species. In congruence with the phylogenomic analysis, Bristol Channel 422 04 is nested within S. subterranea.

Figure 4 Principal Component Analysis (A) and Multi-dimensional scaling (B) of 3,428 SNPs.

The percentage of explained variation is displayed along with the axis for each PC. Lineage (species) are given in different colours. Specimens with ‘intermediate positions’ are highlighted both analyses, indicating potential shared generated variation between specimens.

Population structure, differentiation and summary statistics

The PCA separates the three species across the first two principal components, (which together explain 30.4% of the variance; Fig. 4A). Three individuals stand out, including Bristol Channel 422 04 (labelled as S. josemariobrancoi) which is placed closely with S. subterranea individuals, Bristol Channel 422 05 which occupies an intermediate position between S. westheidei and S. josemariobrancoi, and Lubec 428 02 which is relatively distant from the remaining S. westheidei individuals. The multi-dimensional scaling plot, which is less affected by missing data, separates the species into three distinct clusters (MDS; Fig. 4B). In coherence with the previous analyses, we detect several taxa with intermediate positions: Bristol Channel 422 04 (labelled as S. josemariobrancoi) is closer to the S. subterranea cluster than to the S. josemariobrancoi; Hoernum 169 09 (labelled as S. josemariobrancoi) in an intermediate position between these two species; Bristol Channel 422 05, St. Efflam 401 04, 401 03, 401 05 (all labelled as S. josemariobrancoi) are found in an intermediate position between S. josemariobrancoi and S. westheidei; Lubec 429 02 (labelled as S. westheidei) is also distant from the S. westheidei cluster, being relatively close to Bristol Channel 422 05 (Fig. 4B).

The ADMIXTURE analysis confirms shared genetic signal among species. The most supported cluster size was K = 3 (Fig. S1) and is plotted in Fig. 5. In agreement with the phylogenetic network, the MDS and the PCA, S. westheidei is the species with the least amount of admixture, with only 2 individuals sharing a relatively low degree of ancestry with S. subterranea. A majority of S. subterranea individuals (17 out of 30) share genetic variation with S. josemariobrancoi. S. josemariobrancoi has 5 individuals which are admixed from S. westheidei, and an individual (Bristol 422 04) identified as having a S. subterranea ancestry. Notably, individuals from sympatric areas and belonging to S. josemariobrancoi and S. westheidei (Hausstrand, Musselburgh, Lubec) show no signal of shared ancestry. However, five S. subterranea individuals with shared ancestry belong to two sympatric sites (Hausstrand and Musselburgh). f3 statistics were positive, thus suggesting that the observed patterns of admixture are unlikely to be due to recent admixture (Table 1). Notably, two out of three scenarios retrieved Z scores >3 (threshold used for significance), including S. subterranea and S. josemariobrancoi as source and S. westheidei as target, and S. josemariobrancoi and S. westheidei as sources and S. subterranea as target.

Figure 5 ADMIXTURE analysis of 3,428 SNPs shows shared genetic variation.

Stygocapitella josemariobrancoi, S. subterranea and S. westheidei are plotted consecutively from left to right. Populations and specimen-ids are denoted at the bottom, with sympatric-populations in bold and italics. The cladogram follows the tree topology retrieved in Figs. 2A and 2B.

Table 1 f3 statistics testing for hybridization between lineages.

Each row represents a scenario where two species are the source for admixture, and the third species is the target of hybridization. A f3 statistic, the standard error (SE) and a Z-score value calculated with jackknife is provided for each scenario based.

Source 1	Source 2	Target	f3	SE	Z	
S. subterranea	S. josemariobrancoi	S. westheidei	0.91	0.08	11.424	
S. josemariobrancoi	S. westheidei	S. subterranea	1.09	0.24	4.575	
S. westheidei	S. subterranea	S. josemariobrancoi	0.25	0.10	2.605	

Notably, FST estimates among species are high, thus indicating isolation. Pairwise FST comparisons were lower between S. josemariobrancoi and either of the remaining species: 0.53 for S. josemariobrancoi vs S. subterranea, 0.492 S. josemariobrancoi vs S. westheidei and 0.664 for S. subterranea vs S. westheidei (Table 2).

Table 2 Weir-Cockherham Fst-estimate.

Estimates are provided in the lower part of the table, and the number of individuals included in the pairwise estimation is provided in the upper part of the table.

Species	S. subterranea	S. josemariobrancoi	S. westheidei	
S. subterranea	–	52	48	
S. josemariobrancoi	0.530	–	40	
S. westheidei	0.664	0.492	–	

Summary statistics suggest that populations of S. josemariobrancoi have a higher degree of genetic variation. Waterson’s estimate, S, ranges between 2.11–2.39 in S. subterranea and between 1.81–2.35 in S. westheidei, and between 1.89–6.83 in S. josemariobrancoi, with Bristol Channel (S = 4.92), and Gravesend (S = 6.93; Table 3). This is similar when analysing nucleotide polymorphisms (π), with S. subterranea ranging between 0.002–0.0037, S. westheidei between 0.0021–0.0034, but S. josemariobrancoi ranging between 0.0026–0.0099 again with Gravesend and Bristol Channel as outliers (Gravesend π = 0.0099, Bristol Channel π = 0.0086; Table 3). Interestingly, the sympatric sites do not reveal any signal of higher polymorphism, as it would be expected in scenarios of on-going hybridization. For instance, π = 0.0033 and S = 2.13 for S. subterranea in Musselburgh, and π = 0.0045 and S = 2.49 for S. josemariobrancoi in Musselburgh. Hausstrand, for which we were only able to obtain data for S. josemariobrancoi retrieved a π = 0.0038 and a S = 2.55, while the population Lubec of S. westheidei shows π = 0.0029 and S = 2.15. Tajima’s D excludes the possibility for bottlenecks, as none of the populations exhibits significant Tajima’s D (i.e., values below -2 or above +2; Table 3).

Table 3 Summary statistics for the various analysed populations.

For each site we provide the number of specimens and chromossomes, number of loci considered, S (Waterson’s estimate), the averaged π, and the averaged Tajima’s D. Analysed populations include those with >5 chromossomes, with the exception of Plymouth which had no samples with no missing data.

Species	Site	Number of specimens (chromossomes)	Number of loci analyzed (without missing data)	S(Waterson’s estimate)	Averagedπ	Averaged Tajima’s D	
Stygocapitella subterranea	Ardtoe	7 (14)	122	2.1066	0.0020	−0.1843	
	Glenancross	4 (8)	658	2.1763	0.0029	0.1123	
	Hausstrand	1 (not analyzed)	–		–		
	Ile Callot	3 (6)	1301	2.3912	0.0037	0.1958	
	Keitum	6 (12)	40	2.525	0.0030	−0.1858	
	Little Gruinard	1 (not analyzed)	–	–	–	–	
	Morsum	3 (6)	1923	2.1482	0.0032	0.1326	
	Musselburgh	3 (6)	1166	2.1329	0.0033	0.1571	
	Nairn	2 (not analyzed)	–	–	–	–	
Stygocapitella josemariobrancoi	Bristol Channel	2 (not analyzed)	–	–	–	–	
	Ellenbogen	4 (8)	403	1.6377	0.0024	−0.1993	
	Gravesend	2 (not analyzed)	–	–	–	–	
	Hausstrand	4 (8)	154	2.5519	0.0038	−0.3862	
	Hoernum	3 (6)	261	2.6858	0.0046	−0.0281	
	Lubec	1 (not analyzed)	–	–	–	–	
	Plymouth	0 (not analyzed)	–	–	–	–	
	Musselburgh	3 (6)	414	2.5918	0.0045	−0.0356	
	Saint Efflam	3 (6)	572	3.3969	0.0059	0.1702	
Stygocapitella westheidei	Canoe Beach	5 (10)	577	1.8128	0.0021	−0.3654	
	Lubec	5(10)	89	2.1461	0.0029	−0.1044	
	Reid State Park	5 (10)	519	2.3468	0.0024	−0.5266	
	South Lubec	3 (6)	524	2.0115	0.0034	0.1723	

Simulation of demographic scenarios suggests admixture may have happened in ancestral branches (Fig. 6). All the best supported scenarios are provided in https://github.com/jcerca/Papers/blob/main/Stygocapitella_PeerJ/11_fscScenarios/ and as supplementary. Three of the top four most supported scenarios using likelihood suggest ancient admixture: geographic gene flow (i.e., gene flow between the two ancestral branches and afterwards between S. subterranea and S. josemariobrancoi), ancient gene flow (i.e., gene flow between the two ancestral branches), ancient gene flow and between western lineages (i.e., gene flow between the two ancestral branches and afterwards between S. westheidei and S. josemariobrancoi). The remaining scenario suggests no gene flow. The most supported of these scenarios was the geographic gene flow, which includes coalescent times of 411,402 generations for the first coalescent event (presumably, Stygocapitella has a generation time of a single year), and 21,680,033 for the second coalescent event. The ancient gene flow scenario, which was the second most supported, includes estimates of 585,534 generations for the first coalescent event and 16,749,615 generations for the second coalescent event. The no gene flow scenario is the 3rd most supported scenario, but the first and second coalescent events are suggested to have occurred 451 and 12,834 years ago, being clearly at odds with previous evidence suggesting these lineages diverged millions of years ago (Struck et al., 2017; Cerca et al., 2020a). The fourth most supported scenario, ancient gene flow and gene flow between western lineages, has coalescent times of 3,566,294 and 18,353,555 (Fig. 6). Finally, while the AIC assessment provide slightly different results (Fig. S8), the second and third most supported scenarios are the geographic gene flow and ancient gene flow; being thus in agreement with the likelihood results.

Figure 6 Demographic scenarios considered.

The likelihood of different demographic scenarios is displayed on the Y axis. Based on the estimated phylogeny (Fig. 2), we modelled scenarios for (from left to right): (1) geographic gene flow (gene flow between S. josemariobrancoi and the ancient lineage, and S. josemariobrancoi and S. subterranea); (2) ancient gene flow (gene flow between S. josemariobrancoi and the lineage before the S. subterranea and S. westheidei split); (3) no gene flow at all; (4) ancient gene flow and gene flow between S. subterranea and S. westheidei; (5) gene flow only between S. westheidei and S. subterranea; (6) gene flow in every possible branch; (7) gene flow in sympatric, European linages; (8) gene flow between S. josemariobrancoi and S. westheidei; (9) gene flow between currently existing lineages; (10) ancient gene flow with gene flow between S. josemariobrancoi and S. westheidei.

Discussion

Morphological similarity through extended periods of times, or stasis, has been hypothesized to occur under three possible scenarios underlied by genetics: homogenous genetic variation (due to e.g., ILS, hybridization), genetic constraints (e.g., pleiotropy), and lack of genetic variation (e.g., bottlenecks and founder effects) (Futuyma, 2010). While our sampling design does not account for genetic constraints, we study the evolutionary history of Stygocapitella species seeking to determine signals of loss of genetic variation or shared genetic variance. We find that the three morphologically similar Stygocapitella species herein studied share genetic variation and exclude the possibility of recent bottlenecks or recent admixture. Demographic and admixture analysis reveal signatures of ancestral admixture and incomplete lineage sorting during the divergence of these three species. We discuss the possible implications of these processes to the genomic underpinnings of indistinguishable morphology among cryptic species.

Whole genome amplification and the generation of RADseq data

We show that WGA combined with RAD sequencing may become an important tool for microscopic eukaryote genomics. RADseq library preparation typically requires 200–500 ng of DNA per individual, yet DNA extraction of a single Stygocapitella individual typically yields 10–100 ng of DNA, thus representing a challenge to obtain genome-level data. One potential solution is to pool individuals. However, pooling may not be ideal when dealing with morphologically similar species, especially when they occur in sympatry, as observed in Stygocapitella, since the identification of individuals based solely on morphology may be impossible. Arguably, one of the major advantages of RADseq is to open up population genomics and phylogenomics as approaches for non-model systems at an affordable cost. In such systems, experimental designs may benefit from the inclusion of the largest number of individuals possible, which encompass the whole spectrum of populations or species to determine species boundaries, phylogeography, population structure and the phylogeny. Pooling individuals from different species or populations together may lead to an incorrect inference of the phylogeny when species boundaries are not known, but also skew allelic variation. Consequently, pooling of individuals may result in difficulties during data processing and interpretation, and may require extra efforts such as barcoding of individuals before pooling. In the view of these challenges, we optimized and applied a whole-genome amplification protocol to obtain genome level data, thus confirming its potential for population genomic-inference and phylogenetics (De Medeiros & Farrell, 2018; De Medeiros & Farrell, 2020).

Bottlenecks, meiofaunal dispersal and morphological similarity

The few phylogeographic studies available for meiofaunal organisms have generally detected founder effects and bottlenecks, and have discussed how colonization dynamics may be determined by a series of bottlenecks (Casu & Curini-Galletti, 2006; Derycke et al., 2007; Andrade, Norenburg & Solferini, 2011), however, we find no evidence for bottlenecks in this dataset when using summary statistics and Tajima’s D. Evidence for the prevalence of bottlenecks in meiofauna has been further supported by experimental evidence which showed that colonization of new areas may be characterized by founder effects and bottlenecks, which, in turn, are expected to shape the genetic differentiation of meiofaunal populations (Derycke et al., 2007; Derycke, Backeljau & Moens, 2013). Broadly, this follows hypotheses on marine-invertebrate biogeography which suggests that repeated extinction and recolonization dynamics may be involved in shaping genetic differentiation in populations and species (Andrade, Norenburg & Solferini, 2011; Derycke, Backeljau & Moens, 2013). We find no indication of bottlenecks in Stygocapitella, as suggested by non-significant Tajima’s D values (Table 3), even though that part of the sampling distribution of Stygocapitella included areas which were glaciated only 10,000 years ago (Wares & Cunningham, 2001). Two possibilities may explain this scenario and support our findings. First, the hypothesis that meiofauna disperse through a series of bottlenecks may require more evidence. Indeed, when scoring a total of 752 papers, we could only detect 48 studies focusing on biogeography and 7 on evolutionary biology (including population genetics) (Cerca, Purschke & Struck, 2018), thus suggesting that meiofaunal biogeography is in its early days and more studies are needed for more solid conclusions. This hypothesis may also be at odds with evidence that meiofauna may be, indeed, good dispersers (reviewed in Cerca, Purschke & Struck, 2018). Even if dispersal is carried by only a limited group of individuals, concomitant with the idea of a founder effect and bottleneck colonization, the dispersal of more organisms from ‘source populations’ (multiple waves of dispersal) through time would eventually homogenize genetic variation in newly colonized areas. Second, to the best of our knowledge, this is the first work to focus on population-genomic level data in meiofauna. Typically, works have focused on sequencing a limited combination of mitochondrial and nuclear genes (e.g., Derycke et al., 2005; Derycke et al., 2007; Derycke et al., 2008; Kieneke, Martínez Arbizu & Fontaneto, 2012; Leasi & Norenburg, 2014). While non-recombining data, such as mitochondrial markers, may provide an ideal indication for the occurrence of bottlenecks, these effects should be confirmed on complementary genomic regions. For instance, using the 16S mitochondrial marker, we previously detected single haplotypes in populations of S. subterreanea or S. josemariobrancoi. These populations had a statistically significant Tajima’s D (S. josemariobrancoi individuals from Bristol Channel; S. subterranea individuals from Glenancross) (Cerca et al., 2020b). While this pattern was in conflict with that of the nuclear ITS1 (Cerca et al., 2020a), it is further rejected when using genome-level data, which provides a more comprehensive, and independent, assessment of genomic variance. This suggests that the dispersal-by-bottlenecks idea in meiofauna warrants more data, and that biogeography of meiofauna will benefit from more genomic studies.

Overall, we suggest that morphological similarity in Stygocapitella is unlikely to result from the lack of standing genetic variation due to re-occurring bottlenecks. Under this hypothesis, it is expected that bottlenecks reduce genetic variation, which will in turn limit morphological evolution, thus leading to stasis (Futuyma, 2010). Given the lack of evidence for recent bottlenecks in Stygocapitella, this does not seem plausible. In addition to the evidence for the lack of recent bottlenecks, the fact that the remaining 8 Stygocapitella species live in similar habitats and are distributed throughout the world (Cerca et al., 2020a) indirectly suggests that bottlenecks may not be typical in the evolutionary history of the group.

Evidence for incomplete lineage sorting and admixture

We find clear evidence for shared genetic variation in Stygocapitella. The most conspicuous evidence for this comes from the admixture analysis, which clearly demonstrates admixed populations in the three species (Fig. 5). This evidence is further supported by individuals with intermediate positions in the MDS—a test which is robust to missing data (Fig. 4). However, several evidences do not support a preponderant role of recent admixture. First, we obtained no evidence for admixture when using F-statistics, since we find only positive F-values (Table 1). Second, contrary to the expectation of ongoing gene flow, we do not observe higher levels of heterozygosity in sympatric populations (Lubec in the USA, Musselburgh in Scotland, Hausstrand in Germany; Table 3) where individuals of different species are found in the same sediment sample in close proximity (volume ranging from 50–500 cm3). Third, admixture often generates incongruence between mitochondrial and nuclear markers (Melo-Ferreira et al., 2012; Sloan, Havird & Sharbrough, 2017), which is not seen in single-marker trees (Figs. S2–S5). Fourth, models with exclusive recent admixture are generally poorly supported by the demographic analysis (Fig. 6). In contrast, three out of the four most supported demographic scenarios suggest ancient admixture, and one supported no gene flow at all (Fig. 6). The scenario with no gene flow inferred coalescent times of 451 and 13,834 generations or years (1 generation is expected to be 1 year, Günter Purschke pers. comm)] which are not compatible with estimates of the splitting age of the three Stygocapitella species (∼5–30 million years ago; Cerca et al., 2020b). Given that reduced times of coalescence are a typical signature of simulations that do not account for gene flow, when it has occurred in empirical data (Leaché et al., 2019), it is likely that incomplete lineage sorting alone cannot explain the patterns of shared variation among Stygocapitella species. In other words, the demographic analysis supports a scenario that includes ancient admixture. The three scenarios with ancient admixture vary in the presence or absence of admixture after the second coalescence event (S. subterranea and S. josemariobrancoi): in one scenario, admixture is exclusive to the ancestral branch; in the remaining two, gene flow between S. josemariobrancoi and either S. westheidei or S. subterranea occur. Given the lack of support for on-going gene flow between species by the FST, summary statistics, and F-statistics (Tables 1–3, Figs. S1–S5), admixture may have occurred immediately before or after the speciation event of S. westheidei or S. subterranea, but not in recent times (i.e., the last generations). Furthermore, the occurrence of ancient admixture can affect the inference of recent admixture when not take the phylogeny into account (Malinsky et al., 2018; Ferreira et al., 2020), this may explain the incongruence between some of our analysis. Therefore, while the demographic analysis suggests the occurrence of admixture among S. josemariobrancoi and the other species, future studies are necessary to confidently dissect and determine the role of recent gene flow in the system with independent analyses. For example, these studies will benefit from using whole-genome data to determine whether interspecific divergence in regions of the genome show gene-species tree discordance, thereby dissecting ILS and recent hybridization (Joly, McLenachan & Lockhart, 2009; Giska et al., 2019). Also, the demographic analysis favouring a preponderant role of ancient admixture does not exclude the occurrence of ILS, and the beforementioned approach would also allow to clarify the relative contribution of ILS and gene flow to shared patterns of variation among species. In sum, to the extent that we can speculate, our data suggests that shared genetic variance is more likely explained by an evolutionary history including incomplete lineage sorting and ancient geneflow.

Evidence for ancient admixture or incomplete lineage sorting is further seen in the phylogenomic analysis. Phylogenetic approaches, which seek to reconstruct the evolutionary history of lineages, often fail to resolve the evolutionary history and the ‘true tree-topology’ when the taxa in question have high rates of incomplete lineage sorting or admixture (Kubatko & Degnan, 2007; Degnan & Rosenberg, 2009), but incongruence may also result from tree-building errors, paralogy or horizontal gene transfer (Scornavacca & Galtier, 2017). We discard tree-building errors based on the following evidence. First, individuals with intermediate positions in the PCA and MDS correspond to those causing paraphyly in the trees. Second, when exploring the effects of missing data through (a) labelling the tree with % of missing data (Fig. 2); and (b) constructing a cladogram based only on the shear % of missing data (i.e., UPGMA tree, Fig. S6); we find the placement of intermediate individuals is not guided by missing data. Should missing data determine their placement, we would expect these specimens to nest in close proximity in the UPGMA tree. Finally, in the phylogenetic network, the individuals which are also far removed from the remaining S. josemariobrancoi individuals, occupying central positions or being paraphyletic in the phylogenetic network correspond to those paraphyletic in the tree and in intermediate positions in the PCA and MDS.

In sum, incomplete lineage sorting and ancient hybridization are known to contribute to levels of shared variation among species (Pease et al., 2016; Malinsky et al., 2018; Edelman et al., 2019), even at deep evolutionary levels (Song et al., 2015; Suh, Smeds & Ellegren, 2015). The development of tools which employ the substantial amount of modern genomic data has allowed separating cases of ILS and ancient admixture, showing that ancient hybridization can have a strong impact in the levels of shared variation among species complexes (Malinsky et al., 2018; Li et al., 2019; Taylor & Larson, 2019; Ferreira et al., 2020), even after several million years of divergence (Barth et al., 2020; Suvorov et al., 2020). Future work should employ multispecies-network methods or coalescent simulations (Joly, McLenachan & Lockhart, 2009) to determine the relative role of ILS and ancient admixture. However, it does not seem unlikely that both processes might have thus contributed to levels of shared variation across 5–30 millions of the divergence of Stygocapitella (Cerca et al., 2020a; Cerca et al., 2020b).

Incomplete lineage sorting and morphological similarity

The debate on morphological similarity is slowly shifting from ‘are cryptic species an artefact of systematics?’ to ‘what are the causes underlying morphological similarity?’, following the evidence that speciation is not necessarily accompanied by morphological divergence (Wada, Kameda & Chiba, 2013; Swift, Daglio & Dawson, 2016; Cerca et al., 2020b). We have previously argued that the study of morphological similarity will benefit from predictions, models, and evidence from paleontological stasis (Cerca et al., 2020a), which positions that stasis may result from constraints, selective pressures on physiology and/or behaviour, stabilizing selection, niche conservatism (Hansen & Houle, 2004; Estes & Arnold, 2007; Futuyma, 2010). While similarity in different cryptic species complexes may stem from different causes, morphological similarity in the three studied Stygocapitella species complex is likely associated with homogeneous genetic variation caused by incomplete lineage sorting and ancestral admixture that occurred during the divergence of the complex (Futuyma, 2010). In such a scenario, it is expected that patterns of genetic variation remain similar for the species, thus resulting in the retention of symplesiomorphic morphological states (Futuyma, 2010) and in the deceleration of morphological evolution (Cerca et al., 2020b). In any case, future works using whole-genome data are necessary to, for example, detect if regions affected by incomplete lineage sorting and gene flow are disproportionally enriched for genes that usually contribute for morphological divergence in closely related taxa. These works should also employ more variant-level data to confirm the patterns herein obtained.

Conclusions

The increasing discovery of cryptic species has led to heated debates in systematics, mostly lacking an integration in an evolutionary framework. Here, we tested the hypotheses that morphological similarity may own to reduced genetic variation (bottlenecks, founder effects), recent admixture (shared genetic variation), or incomplete lineage sorting and ancient admixture. We found that morphological similarity in the three morphologically similar Stygocapitella species may own to incomplete lineage sorting or ancient admixture underlying shared genetic variation. Future works should focus on understanding whether reduced genetic variation or shared genetic variation underlies morphological similarity in other systems.

Supplemental Information

Supplemental Information 1 Studied species, sites including GPS coordinates, population map IDs (as used in stacks) and number of specimens used in this study

Sites where multiple species were found (i.e. sympatric sites) are given in bold.

Click here for additional data file.

Supplemental Information 2 Specimens used in this study

For each specimen we provide a sampling code, the collection site, a sampling code and the NCBI information for COI, 16S, 18S, ITS1. The column “Present in the final dataset” shows whether the specimen was removed due to ¿90% missing data, as shown in the final column.

Click here for additional data file.

Figure S1 Cross validation error for the Admixture analysis

The cross-validation error is provided in the Y axis and the different K in the X axis.

Click here for additional data file.

Figure S2 16S mitochondrial marker tree

Bootstrap support for the four species is provided above the branches. Stygocapitella zecae is added as outgroup.

Click here for additional data file.

Figure S3 18S nuclear marker tree

Bootstrap support for the four species is provided above the branches. Stygocapitella zecae is added as outgroup.

Click here for additional data file.

Figure S4 COI mitochondrial marker tree

Bootstrap support for the four species is provided above the branches. Stygocapitella zecae is added as outgroup.

Click here for additional data file.

Figure S5 ITS1 nuclear marker tree

Bootstrap support for the four species is provided above the branches. Stygocapitella zecae is added as outgroup.

Click here for additional data file.

Figrue S6 UPGMA Tree reconstructed from % of missing data

Specimens from different species are coloured differently.

Click here for additional data file.

Figure S7 Phylogenomic tree based on 4,737 RADseq loci

Rad-seq alleles (0 and 1) were converted into a consensus sequence. Bootstrap support is provided for the main branches. Coloration follows species with blue representing Stygocapitella subterranea, green representing S. josemariobrancoi, and orange S. westheidei.

Click here for additional data file.

Figure S8 AIC-evaluation of the demographic scenarios

Different models are depicted in the X axis and have different colours, AIC values are given in the Y axis. Species names are reduced with ‘s’ representing Stygocapitella subterranea, ‘i’ representing Stygocapitella josemariobrancoi, and ‘w’ representing Stygocapitella westheidei.

Click here for additional data file.

JC is grateful to Tim Worsfold, Andy Mackie, Henning Reiss, Lis Jørgensen for laboratory space in the UK and Norway. We thank Audun Schrøder-Nielsen and Lisbeth Thorbek for assistance in laboratory work, and Inês Modesta for fieldwork support. We are grateful to Diego Fontaneto, Gerardo Perez-Ponce de Leon and an anonymous reviewer for their comments, suggestions and critiques, which have led to the substantial improvement of this manuscript. We acknowledge the use of Norwegian national infrastructure for high-performance computing and storage via the projects NN9408K and NS9408K, respectively. In concordance and support of transparent science, the raw data associated with this paper is publicly available in the European Nucleotide Archive (Project PRJEB40223). This is NHM genomics laboratory contribution 22.

Additional Information and Declarations

Competing Interests

Author Contributions

Data Availability

The authors declare there are no competing interests.

José Cerca conceived and designed the experiments, performed the experiments, analyzed the data, prepared figures and/or tables, authored or reviewed drafts of the paper, and approved the final draft.

Angel Rivera-Colon analyzed the data, authored or reviewed drafts of the paper, and approved the final draft.

Mafalda Ferreira, Mark Ravinet and Julian Catchen analyzed the data, authored or reviewed drafts of the paper, and approved the final draft.

Michael Nowak conceived and designed the experiments, authored or reviewed drafts of the paper, and approved the final draft.

Torsten Struck conceived and designed the experiments, performed the experiments, analyzed the data, authored or reviewed drafts of the paper, and approved the final draft.

The following information was supplied regarding data availability:

Code is available at Github:

https://www.ebi.ac.uk/ena/browser/view/PRJEB40223.

The data is available in the ENA: PRJEB40223.

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
