# Peer review of "Incomplete lineage sorting and ancient admixture, and speciation without morphological change in ghost-worm cryptic species"

_PeerJ, doi:10.7717/peerj.10896_

## Round 0.1 · original submission · Major Revisions

Dear Dr. Cerca and colleagues:

Thanks for submitting your manuscript to PeerJ. I have now received three independent reviews of your work, and as you will see, the reviewers raised some concerns about the research. Despite this, these reviewers are optimistic about your work and the potential impact it will have on research studying ghost-worms and cryptic species complexes. Thus, I encourage you to revise your manuscript, accordingly, taking into account all of the concerns raised by both reviewers.

There are many suggestions, which I am sure will greatly improve your manuscript once addressed.

Importantly, please consider comments about the rationale. Also, ensure that the phylogenetic methods are robust and account for missing data and paralogy.

I look forward to seeing your revision, and thanks again for submitting your work to PeerJ.

Good luck with your revision,

-joe

·

Basic reporting

The manuscript represents one of the fist comprehensive studies of gene flow between cryptic taxa in a species complex of marine microscopic animals. Meiofauna is one of the groups of animals where DNA sequences are often used to delimit taxonomic units, and this study goes a step further in attempting to understand the level and timing of gene flow to explain morphological similarity. The analyses are appropriate and provide supported results for the inference. I wish that such approach could become the standard in meiofauna and I congratulate the authors for their work.

Experimental design

The experimental design is fine

Validity of the findings

the findings are highly novel and relevant.

Additional comments

I have a series of comments to improve the strength of the message, removing weaknesses or ambiguities. These may be just subjective opinions, but consider that also other readers may have the same doubts.

1. What I do not find completely convincing is the connection between past gene flow and morphological similarity in cryptic species. The fact that gene flow exists does not explain the reason for morphological similarity. There are several examples among plants and animals, even meiofauna, where extensive gene flow and hybridisation do not preclude the possibility to identify species morphologically. This part of the rationale should be rephrased to provide convincing statements, or should be removed. I would opt for its removal, given that it does not provide more interest to the results and it only makes the rationale less convincing.

2. Another part of the rationale that is not supported is in the first paragraph of the discussion (and in parts of the introduction), where the search for the reason for the morphological similarity is not due to bottlenecks or recent admixture, and thus it is stated to be “in line with an incomplete lineage sorting scenario”. The ILS scenario is thus supported in the absence of the other causes, omitting other potential explanations. The presence of ILS is not even demonstrated, because it seems that hybridisation could be the reason for gene sharing. Thus, if hybridization exists, no ILS can be supported. Maybe a test to support whether ILS or hybridization is present should be included. Something along the line of the statistical rationale used by Joly et al (https://www.journals.uchicago.edu/doi/full/10.1086/600082) may do the job. In any case, even after such test, the connection with morphological similarity is weak. Only a comparison with the level of gene flow between closely related species groups of taxa that are morphologically distinguishable and diverging at the same time could be used to identify the reason for morphological similarity in cryptic taxa.

3. Overall, the idea is great, the data and the analyses surely worth, but the rationale to fit the results in the idea of searching for a reason for morphological similarity weak, and not necessary.

Other issues
4. The idea of cryptic species being taxonomic artefacts is reported in the abstract and in the introduction without any further explanation. My suggestion is to (i) expand the idea and clearly explain it in the introduction, given that most readers will disagree on it, and to (ii) remove it from the abstract, where it may represent an ambiguous weakness for several readers.

5. Also the idea that cryptic species are overlooked, on line 46, is not unambiguous. If anything, most taxonomic studies nowadays deal with DNA sequences and make integrative taxonomy the rule and not the exception. I understand that this is only my subjective opinion, based on the organisms I work with; yet, there is no reason to upset some readers with unsupported statements and I would suggest to remove the phrase ‘yet overlooked’ on line 46.

6. The nomenclature of the species complex is not always clear. For example, on line 89, it is reported as “The Stygocapitella species cryptic complex”, which does not make much sense, because (i) it does not report the name of the species complex but only of the genus, and (ii) the order “species cryptic complex” is not understandable. The same problem of unclear nomenclature appears already in the title and in the abstract: a complex of cryptic species cannot be identified by the name of a genus. One genus may contain several easily distinguishable species and also several species complexes, which are all named individually usually by using the name of one of the species in the complex. The problem is clearly seen in the sentence on line 89, where 11 species with 4 morphotypes are mentioned, revealing that not only one, but 4 complexes of cryptic species, which cannot be distinguished using morphology alone, exist in the genus. Please, remove this ambiguity in the terminology and clarify what the complexes are.

7. The hypotheses at the end of the introduction are unclear and unsupported. H1 bottlenecks and founder effects should not only provide morphological similarity; bottlenecks and founder effects, by reducing genetic variability within each population of each species, maximise differences between populations and between species, which is the opposite of what is stated. It is true that bottlenecks reduce genetic variability, as stated in the discussion on line 512, but bottlenecks are one of the traditionally accepted mechanisms to have genetic diversity between populations, quickly leading to speciation, again in the opposite direction of what is assumed in the hypothesis. Think of the bottleneck and founder effects on islands, quickly leading to morphological differences between the island and the source populations. Please, clarify the rationale for this hypothesis. H2 implies that hybridisation makes species more similar, but there are lots of examples of complexes of hybridising species that can still be distinguished morphologically, for example in plants but also in other microscopic aanimals where evidence of mitonuclear discordance has been found. H3 could be OK, but the fact that ILS is present should be clearly tested and not assumed.

8. On line 86-87, the sentence “Very few biogeographic and systematic studies, let alone evolutionary studies, have thus contributed to the discussion” is then followed by a list of 6 references (several from the authors), making the sentence not so supported. A list of 6 references cannot be considered to represent “very few” studies, considering that several other studies have not been mentioned. Especially in meiofauna, the use of DNA taxonomy in the study of biodiversity in all its aspects (including biogeography and evolution) has a long history and several readers may not agree on the sentence.

9. Given that the study is on species, it would be better to use zoological nomenclature according to the strictest rules in taxonomy, reporting species authorship at the first citation of a species in the main text, taking care of reporting the correct use of parentheses.

10. The chapter starting with the title on line 159 on “Species delimitation” does not describe any method of species delimitation but only the “multi-marker phylogeny”. Either change the title or include a description of the methods that are used for species delimitation.

11. In the same chapter, a sentence should state whether ITS was present with double peaks or not, and how the double peaks were treated. Check this method for double peaks in ITS in species delimitation: https://besjournals.onlinelibrary.wiley.com/doi/abs/10.1111/2041-210X.13454. I am not suggesting to use haploweb, but I only want to point out that the reader may need to know what happened to the double peaks in ITS1.

12. Line 359: ‘taxa’ does not seem to be the correct term, as the idea refers to ‘individuals’

13. The long discussion on pooling individuals for RADseq seems out of place. Please, shorten it considerably. It is already convincing that the study managed to get good genomic data from single animals through WGA. Also the caveats due to WGA should be shortened, and especially moved at the end of the discussion. It would be much better first to discuss the biological implications of the results, and then only later introduce the potential problems. Starting the discussion with problems seems to distract the reader. Just a minor point: WGA works also for rotifers, which are much smaller than the analysed anellids (https://www.sciencedirect.com/science/article/pii/S0960982216000816; https://journals.plos.org/plosbiology/article?id=10.1371/journal.pbio.2004830).

14. The discussion in the part on line 521 is weak. ILS and hybridisation can be distinguished statistically (https://www.journals.uchicago.edu/doi/full/10.1086/600082) and the manuscript remains too ambiguous on this point, stating that admixture could be due to one of the two processes, but then without testing which is more likely.

15. In the same part of the discussion, the speculation on sympatric and non sympatric populations is also weak: if the estimates are correct, gene flow can happen every thousands of years, meaning that animals from different populations have the time to move from one population to the other.

16. The selection of references is a bit too much biased towards the production of the authors. Whereas I would say that it is a good option for the discussion, having too many references from the authors in the introduction may leave the impression that only the authors care about this topic, with no broad interest. This is not true, and several studies on cryptic species, their origin, and their biogeography exist in the meiofauna, other than in anellids.

17. The choice of colours representing the three species in the figures is not the best one: two of the three species will not be clearly seen by several readers with colour vision impediments (green and orange may be mixed up). Choose a set of three easily distinguishable colours to be more inclusive.

18. The English language is rather good, but please consider a careful revision, in order to remove very minor problems such as:
- Single comma between subject and verb (e.g. on line 42).
- No hyphens are allowed for adverbs ending with –ly (e.g. on line 49, 61, 62, etc.); this is called a hypercorrection, because no ambiguity is possible with such adverbs and thus the hyphen is redundant. The text is already ambiguous in the use of such hyphen, because in some instances the adverbs are already reported without the hyphen (e.g. line 91).
- Hyphens are not used consistently also in other phrases, for example “missing-data” on line 274 should not have the hyphen. Please, carefully check the use of all the hyphens in the text.
- The term ‘specimen’ is used ambiguously: strictly speaking, ‘specimen’ is a representative individual or part of it, stored in a collection, whereas in the manuscript it is used with the meaning of a single individual. I guess that the animal is completely destroyed by the DNA extraction, and thus the ‘specimen’ is lost and cannot be considered a specimen anymore. In addition, the phrase ‘individual specimen’ is a tautology. The misleading use of ‘specimen’ is only a minor problem, but for entomologists and other readers who work in museums the manuscript will seem weird. Thus, there is nothing to gain by using ‘specimen’ instead of ‘individual’, which will make all readers happy.
- ‘a loci has’ should be ‘a locus has’.
- ‘These statistics consists’ should be ‘These statistics consist’.
- ‘which’ is used for both restrictive and non-restrictive clauses, but at least one instance of the use of ‘that’ instead of ‘which’ are present (e.g. line 485) and should be removed for consistency.
- On line 368, ‘the least amount’ should be ‘the lowest amount’, because amount is countable.

I hope that my comments could be useful to strengthen the impact of the message of the study.
Yours sincerely,
Diego Fontaneto

·

Basic reporting

I think this is an extraordinary contribution to our understanding of the factors that determine the existence of cryptic species, and the current debate of the increasing discovery and the lack of integration into an evolutionary framework as discussed by the authors.
In my opinion the paper covers the most relevant literature published on the subject, it is very well-written and structured and all the hypothesis are very well tested and discussed. No ambiguity I can see from reading this nice piece of work .
On lines 56-64 author present idean on how the discovery of cryptic species is important in terms of biological systematics, but also in other fields in terms of their practical impact, I would suggest to use the example of parasitic organisms, where this fact has been clearly shown at least in two studies (Pérez-Ponce de León and Nadler, 2010 Journal of Parasitology 96, 453-464, and Nadler and Pérez-Ponce de León 2011 Parasitology 138, 1688-1709. .

It is mentioned in the introduction that the genus contain 11 species, although in WoRMS (World Register of Marine Species), only 8 are reported.

Experimental design

The methodological approach to test the hypothesis that morphological similarity among three species of the polychaet Stygocapitella is the result of either bottlenecks, recent admixture or incomplete lineage sorting is sound and actually it is on the state-of-the-art since it uses whole genome data and a vast array of methods currently available.
In my opinion, the question is fully addressed and the conclusions are very clear.

I have a question regarding the use of independent molecular markers. It is mentioned in the results section lns 286-294 that authors compiled a dataset of 4,147 bp, for 4 molecular markers (COI, 16S, ITS1, 18S), and the phylogenetic tree is the result of the concatenated analysis (Fig. 2A), although the sample size for each of them is quite variable, did the authors check for the reliability of presenting a concatenated analysis of the 4 markers?. I think that part of the methodology is very short and not fully explained.

Validity of the findings

The approach followed by this paper is novel, and points to the use of genome data to fully understand the factors that determine the occurrence of cryptic species in natural systems, including the 11 potential causes outlined by authors on lns 71-82 of this paper.
In my opinion, conclusions are clear and open a clear path to demonstrate with other empirical studies that the processes underlying the morphological similarity among the 3 species of Stygocapitella studied in this work. The analyses left some room for speculation, which in my opinion is well justified.

Additional comments

Very nice piece of work!.

Reviewer 3 ·

Basic reporting

In this contribution, Cerca et al study the population genetics of 3 cryptic and partially sympatric species of the meiofaunal worm Stygocapitella. Based on previous work with a limited number of markers, the 3 lineages were recognized and here they evaluate whether the morphological similarity is a result of recent bottlenecks, continuous gene flow or incomplete lineage sorting. The paper is well-written and figures are informative. The DNA data deposition statement includes raw Illumina reads but not Sanger sequences (maybe it could mention they are in supp table 2 for easy access?). Code is not provided in the current version, which prevented interpretation of some results (see Experimental design)

Given the general lack of knowledge on the diversification of meiofauna and the huge diversity of these animals, this paper is a relevant advance showing how speciation does not necessarily lead to morphological differentiation, and attempting to rule out some of the hypotheses explaining that. The extensive geographical sampling and transparency in presentation of results are also noteworthy. Another important outcome of the study is the presentation of a methodology to work with challenging samples represented by these small animals: some of the reasons for the lack of studies are precisely the difficulty in sampling and applying standard molecular protocols to these organisms.

I believe, however, that some of the results presented in the paper could be artifactual, and more work could be done to demonstrate their validity before publishing. Specifically, despite the efforts to mitigate these problems, missing data could still be affecting the results, the interpretation of coalescent models is problematic, and the data could be further filtered to remove artifacts arising, for example, from paralogy. I believe that rejection of hypotheses 1 and 2 (bottleneck and introgression) could be due to artifacts, and therefore more analyses are needed to demonstrate that this is not the case (or to review the conclusions, if it is). That said, I also believe that all problems can be addressed with the data available, and below I make suggestions for each point.

Experimental design

The research questions are well-defined, but I believe their framing could be improved. Specifically, the end of the introduction seems to suggest that one of the three hypotheses is the cause for cryptic species, when earlier the authors list more processes that cannot be tested with data in hand (for example, stabilizing selection on phenotypes). It is well possible that the three species have not undergone bottlenecks, do not hybridize and do not retain ancestral polymorphisms and yet are morphologically similar. Similarly, finding that ILS exists in this system does not imply that it is responsible for the morphological stasis and other, untested, processes are not occurring (such as stabilizing selection). The authors could acknowledge that more explicitly.

Methods are generally described in detail, but in several instances the authors refer important information to other papers. In some instances (specific comments in minor suggestions below), relevant information could be repeated in this paper. Below I make a few comments on the major points in which methodological choices can impact results:

The parameter files for fastsimcoal are not provided, nor any description of the parameters used. This is important, because the authors use the absolute number of generations to species divergence to rule out some of the coalescent scenarios. It is somewhat unfortunate that fastsimcoal works with absolute numbers instead of coalescent units, since inference of absolute effective population sizes and divergence times is only reliable if there are reliable mutation rates. Since the mutation rate used (and a justification for its choice) is not provided, I believe that direct interpretation of coalescent times in generations is problematic. Therefore, I find the conclusion that there is no gene flow between species unconvincing when models with gene flow are clearly well supported. Moreover, since both ancient gene flow and recent gene flow between s-w were supported, maybe it would be important to add a model with these two migration parameters simultaneously. Finally, it is not clear whether gene flows assumed to be symmetric. If it is highly asymmetric, and assumed to be symmetric, one could be missing scenarios of gene flow with very strong support from the data.

Another important fact for interpretation of results and not thoroughly discussed in text is missing data in the Sanger dataset used to evaluate species identities. It seems that for many samples the data is heavily biased to the two mitocondrial genes. The authors refer to previous work to say that there is no discordance between genes from the Sanger sequencing dataset, but it is not clear if this previous work included the few samples with discordance between datasets found here. Could it be that the labelling of species is only reflecting mitocondrial lineages? In that case, it is possible that rare introgression affecting only a small part of the genome (such as mtDNA) is driving observed patterns, at least in part. Maybe showing trees for each gene, highlighting samples placed "incorrectly" in the concatenated SNP dataset would help in understanding patterns.

Despite all the effort to minimize missing data, it seems that there might be still substantial missing data for some individuals. A table with amount of missing data per sample could help to visualize that. Fig 2A shows a typical pattern when adegenet is used in datasets with large amounts of missing data: samples with large amounts of missing data are pulled to the middle, obscuring patterns. This is because adegenet uses a very simple rule for imputing missing data: replacing these with average allele frequencies. This may be problematic in a sample with substantial missing data (close to 50%, for example), and the differences between MDS (which I assume ignores pairwise missing data) and PCA indicate that this may be a problem.
See, for example, the following reference:
Larson, WA, Isermann, DA, Feiner, ZS. Incomplete bioinformatic filtering and inadequate age and growth analysis lead to an incorrect inference of harvested‐induced changes. Evol Appl. 2020; 00: 1– 12. https://doi.org/10.1111/eva.13122
Moreover, as also discussed in the reference above, it is possible that the dataset includes problems such as paralogy, which can further obscure patterns and affect inferences. The text does not mention a filter for paralogy, for example.
To mitigate both problems, I recommend using genotype likelihoods instead of called genotypes for doing the PCA. This is implemented in PCAngsd:
Meisner J & Albrechtsen A. 2018. Inferring population structure and admixture proportions in low-depth NGS data. Genetics 210: 719–731.
The same program also provides a method for testing sites for Hardy-Weinberg equilibrium while considering population structure, which is a powerful method to remove artifacts such as paralogy:
Meisner J & Albrechtsen A. 2019. Testing for Hardy–Weinberg equilibrium in structured populations using genotype or low-depth next generation sequencing data. Mol. Ecol. Resour. 19: 1144–1152.
Implementing these would require some changes to the workflow in the paper, notably not using genotypes called by stacks but rather using stacks loci only to produce a reference "genome" onto which raw reads are mapped and then used as input to ANGSD (http://www.popgen.dk/angsd/index.php/ANGSD) followed by PCAngsd. Given the nature of the dataset, I believe it is important to use a method less sensitive to missing data. The usage of genotype likelihoods would decrease the amount of missing data by allowing lower-coverage loci, the PCA would be estimated without the bias mentioned above and problematic loci would be filtered. This might have significant consequences for results presented.

A final point that I believe deserves attention is the choice to filter SNPs with minimum allele frequency below 0.05. Given the sample sizes used here, it would be possible to detect alleles with frequencies lower than that, an I am wondering if ignoring low-frequency alleles has an effect on statistics such as Tajima's D. If this is the case, the conclusion that populations did not experience bottlenecks could also be an artifact. I would recommend testing the effects of a lower theshold, perhaps 1/N, where N is the number of samples sequenced.

Validity of the findings

The discussion is well-written. However, given the concerns raised in Experimental Design above, I believe that a careful discussion of the validity of the findings can only be done after concerns about artifacts are addressed.

Additional comments

Minor suggestions

throughout the text: species names are not italicized

line 110 - I believe "Maps" is not the last name of "Google Maps", please revise this reference
line 180 - even though a reference on best practices is cited, the authors could briefly comment on which criteria were used to select -M 3 and -n 3. Why was this considered the best combination?
line 190 - what is PeerJ policy for citing unpublished data?
line 450 - "stochastic biases" - I find this construction confusing, not sure if by this the authors mean random or systematic errors.
Supplementary Table 2 - The legend mentions that "bad apples" are defined in the main text, but no reference is made there.

Fig 2 - what is A0, A1? Are these the concatenation of the two alleles? If this is the case, since RAD loci are generally unlinked, I do not think that considering them as separate terminals makes much sense. Maybe using the consensus for each sample?
Additionally, it is not clear what "shared pairwise data" means. Is this the average number of loci shared with other samples? Why is it different for A0 and A1 for a given sample?

fig 6 - The figure is missing y axis label. Moreover, is this really likelihood, or is this AIC? I believe this may be the AIC, since one would expect the likelihood to increase with more model parameters.

---

## Round 0.2 · Minor Revisions

Dear Dr. Cerca and colleagues:

Thanks for revising your manuscript. The reviewers are very satisfied with your revision (as am I). Great! However, there are a couple issues to still address and a few minor edits to make. Please address these ASAP so we may move towards acceptance of your work.

Best,

-joe

·

Basic reporting

no comment

Experimental design

no comment

Validity of the findings

no comment

Additional comments

The authors did a great job in addressing all the comments of the reviewers, not only mine. Well done. I am sure it will be an interesting paper for several readers.

·

Basic reporting

No further comments

Experimental design

No further comments

Validity of the findings

No further comments

Additional comments

I am very please with the response to my queries. No further comments.

Reviewer 3 ·

Basic reporting

No comment

Experimental design

No comment

Validity of the findings

No comment

Additional comments

There was considerable improvement to the manuscript in response to reviewers' concerns and I believe it is much better now. That said, the paper could benefit from revision of a few points. Here I start by making some comments that have been apparently overlooked during the first round of reviews and then I make a few suggestions based on the current version of the manuscript.
One of them was a problem already in the first version (model comparison by likelihoods), but the figure in the first version had an unlabelled axis and it was unclear whether this was the case. Now that it is clear, I suggest to use the likelihoods to calculate AIC instead of comparing models by likelihood directly. AIC scores should be comparable across models, but likelihood is not due to the different number of parameters. If anything, I believe this will make the main argument stronger, decreasing support for models including migration in relation to models with migration.

Before pointing out parts to be revised, I would like to commend the authors for the careful revision. While it is clear that great care was taken to address major comments, it seems some of minor issues that I pointed out in the first round were neglected, so I rephrase them here:

1 - I still do not understand why samples are split in A0 and A1 in Figure 2 if alleles are unphased. Whether an allele is marked as "0" or "1" is arbitrary because of the lack of phasing, and therefore grouping them this way makes no sense. I would think that a phylogeny based on the consensus sequence for each individual would be more appropriate.

2 - "Maps, Google" is still in the references. I would suggest reviewing references more broadly prior to publication. For example, there are two preprints cited and by now they might have been published already.

3 - "bad apple" is still mentioned in Supp. Table 2 but not defined in text. In the description of the method to minimize missing data one can guess these are the samples with >45% missing data that have been removed, but since the term "bad apples" is used in the supplement it should also be explicitly defined in the main text (or removed from the supplement).

In addition to those, I have a few comments on this version of the manuscript.

1 - Not sure if I am the one who failed here, but I could not find and download fastsimcoal files that should be in the supplement. Please make sure they are provided in the final version (e. g. together with code) to ensure reproducibility.

2 - While the authors disclosed the mutation rate used in the response to reviewers, it is not mentioned in the text. This is important, since the number of generations estimated and discussed depends on mutation rate. For example, the authors mention that species of Stygocapitella have one generation per year, which is as important as mutation rate to interpret how this relates to actual time scales. I would suggest adding it as a short sentence to methods, not only the supplement.

3 - While in the response authors mention not using a minimum allele frequency criterion for estimation of Tajima's D, it is not clear in the text when a maf was used or not. The only time maf is mentioned is in line 216, and it seems implicit that this criterion applies to all downstream analyses. If this is not the case (as seems to be from the response to reviewer's comments), it should be stated explicitly.

4 - Now that it is clear that the y axis in Figure 6 is likelihood, this reveals a problem. Model comparison in a likelihood framework needs to take into account the number of parameters, since a simpler model is preferable to a more complex one if they have the same likelihood. There are different ways of accounting for number of parameters, but the Akaike Information Criteria (AIC) is one of them and widely used in the context of fastsimcoal. AIC = 2*k - 2*L, where k is the number of free parameters (i. e. the number of parameters in the *.est file of fastsimcoal) and L is the log-likelihood. After calculating the AIC for each model, these can be directly compared and the best model should have the lower AIC. There are several tutorials available, I found one of them here, for example:https://speciationgenomics.github.io/fastsimcoal2/
I suspect that after comparing models by the AIC the model with no migration will be favored in relation to others, making the argument in the paper stronger.

5 - line 280, "module as asymmetric" should be "modelled as asymmetric".

6 - There are weird characters following numbers in the paragraph starting in line 294

---

## Round 0.3 · accepted · Accept

Dear Dr. Cerca and colleagues:
Thanks for revising your manuscript based on the concerns raised by the reviewer. I now believe that your manuscript is suitable for publication. Congratulations! I look forward to seeing this work in print, and I anticipate it being an important resource for groups studying ghost-worms and cryptic species complexes. Thanks again for choosing PeerJ to publish such important work.

Best,

-joe

---

## Author Rebuttal · Round 0.3

Editor comments (Joseph Gillespie)
MINOR REVISIONS
Dear Dr. Cerca and colleagues:

Thanks for revising your manuscript. The reviewers are very satisfied with your revision (as am I). Great! However, there are a couple issues to still address and a few minor edits to make. Please address these ASAP so we may move towards acceptance of your work.

Best,

-joe

Dear Joe,

Thank you for your encouraging words. We have now changed the manuscript to meet with the reviewer's comments. This specifically includes:

1. We have included an AIC assessment of the demographic analysis, as argued by #R3. The re-running of the analysis and likelihood, together with AIC, made us obtain slightly different results. Specifically, the models do not reject ancient admixture. We have accommodated these changes by modifying a paragraph in the discussion.
2. We have included a consensus tree in the supplementary, as suggested by #R3.
3. We have established a github page with all source code for this paper. Please find it here: https://github.com/jcerca/Papers/tree/main/Stygocapitella_PeerJ

We have added a note of gratitude in the acknowledgments to the three reviewers. "We are grateful to Diego Fontaneto, Gerardo Perez-Ponce de Leon and an anonymous reviewer for their comments, suggestions and critiques, which have led to the substantial improvement of this manuscript. "

Below we provide a point-by-point answer to the concerns and suggestions raised by #R3, the only reviewer pointing out issues in this round of review.

Thank you for editing our manuscript.

**Reviewer 3 (Anonymous)**

**Basic reporting**
No comment

**Experimental design**
No comment

**Validity of the findings**
No comment

**Comments for the Author**

There was considerable improvement to the manuscript in response to reviewers' concerns and I believe it is much better now. That said, the paper could benefit from revision of a few points. Here I start by making some comments that have been apparently overlooked during the first round of reviews and then I make a few suggestions based on the current version of the manuscript.

One of them was a problem already in the first version (model comparison by likelihoods), but the figure in the first version had an unlabelled axis and it was unclear whether this was the case. Now that it is clear, I suggest to use the likelihoods to calculate AIC instead of comparing models by likelihood directly. AIC scores should be comparable across models, but likelihood is not due to the different number of parameters. If anything, I believe this will make the main argument stronger, decreasing support for models including migration in relation to models with migration.

Before pointing out parts to be revised, I would like to commend the authors for the careful revision. While it is clear that great care was taken to address major comments, it seems some of minor issues that I pointed out in the first round were neglected, so I rephrase them here:

We are particularly grateful to #R3 for their attentive read. The reviewer's suggestions have (yet again!) contributed to a substantial improvement of our manuscript. We are also grateful for the kind words of support. Below we answer point-by-point the reviewer's concerns.

1 - I still do not understand why samples are split in A0 and A1 in Figure 2 if alleles are unphased. Whether an allele is marked as "0" or "1" is arbitrary because of the lack of phasing, and therefore grouping them this way makes no sense. I would think that a phylogeny based on the consensus sequence for each individual would be more appropriate.

Thank you for this comment. We have now added this analysis as Suppl. Figure 7:

[Figure]

"**Supplementary Figure 7.** Phylogenomic tree based on 4,737 RADseq loci, where alleles (0 and 1) were turned into a consensus sequence. Bootstrap support is provided for the main branches. Coloration follows species with blue representing Stygocapitella subterranea, green representing S. josemariobrancoi, and orange S. westheidei."

In the main text we have introduced this analysis by including the following text:

**M&M:** "Additionally, since RADseq loci (represented by allele 0 and allele 1) are not phased and since the labelling of 0 and 1 are arbitrary, we obtained a consensus sequence for each individual. This was done by running the consambig module included in the EMBOSS pipeline (Rice et al. 2000)."

**Results:** "Finally, the generated phylogenomic consensus tree shows a similar topology to the that in Figure 2 (Supplementary Figure 7). The three samples causing paraphyly of the lineages in the phylogenomic tree are placed within S. subterranea (Bristol Channel 422 04), as sister to the lineage S. josemariobrancoi and S. westheidei (St. Efflam 401 03), and as the first branch of S. josemariobrancoi (Bristol Channel 422 05)."

2 - "Maps, Google" is still in the references. I would suggest reviewing references more broadly prior to publication. For example, there are two preprints cited and by now they might have been published already.

We have now removed he google maps reference and have added the two published papers, replacing the bioRxiv reference:

"Ferreira MS, Jones MR, Callahan CM, et al (2020) The legacy of recurrent introgression during the radiation of hares. Syst Biol"

"de Medeiros BAS, Farrell BD (2020) Evaluating insect-host interactions as a driver of species divergence in palm flower weevils. Commun Biol. https://doi.org/10.1038/s42003-020-01482-3"

3 - "bad apple" is still mentioned in Supp. Table 2 but not defined in text. In the description of the method to minimize missing data one can guess these are the samples with >45% missing data that have been removed, but since the term "bad apples" is used in the supplement it should also be explicitly defined in the main text (or removed from the supplement).

Thank you for this comment. We have now corrected this inconsistency by removing the bad apple reference. In the summary, it reads:

"**Supplementary Table 2** Specimens used in this study. For each specimen we provide a sampling code, the collection site, a sampling code and the NCBI information for COI, 16S, 18S, ITS1. The column "Present in the final dataset" shows whether the specimen was removed due to >90% missing data, as shown in the final column."

In addition to those, I have a few comments on this version of the manuscript.

1 - Not sure if I am the one who failed here, but I could not find and download fastsimcoal files that should be in the supplement. Please make sure they are provided in the final version (e. g. together with code) to ensure reproducibility.

Thank you for this comment. The files are available with the review package, and have now been included in the following github page: https://github.com/jcerca/Papers/tree/main/Stygocapitella_PeerJ

2 - While the authors disclosed the mutation rate used in the response to reviewers, it is not mentioned in the text. This is important, since the number of generations estimated and discussed depends on mutation rate. For example, the authors mention that species of Stygocapitella have one generation per year, which is as important as mutation rate to interpret how this relates to actual time scales. I would suggest adding it as a short sentence to methods, not only the supplement.

Thank you for this comment. We have now updated the text on the manuscript:

"When included in the model, gene flow was moduled as asymmetric. Each model was run 10,000 times with an assumed mutation rate of 1.2e-8, and the best fitting scenario was evaluated using likelihood, by running it 100 times."

3 - While in the response authors mention not using a minimum allele frequency criterion for estimation of Tajima's D, it is not clear in the text when a maf was used or not. The only time maf is mentioned is in line 216, and it seems implicit that this criterion applies to all

downstream analyses. If this is not the case (as seems to be from the response to reviewer's comments), it should be stated explicitly.

We have now made it more explicit that we did not account for minimum allele frequency in this analysis. In specific, on line 258 it reads (addition in bold):
"To gauge population-level patterns and diversity, we selected loci from the all-sites dataset without missing-data at the population-level and estimated summary statistics including nucleotide diversity (π), Waterson's estimator of genetic diversity (S) and Tajima's D using DNAsp v6 (Rozas et al. 2017). **This dataset was not pruned for minimum allele frequency.** "

4 - Now that it is clear that the y axis in Figure 6 is likelihood, this reveals a problem. Model comparison in a likelihood framework needs to take into account the number of parameters, since a simpler model is preferable to a more complex one if they have the same likelihood. There are different ways of accounting for number of parameters, but the Akaike Information Criteria (AIC) is one of them and widely used in the context of fastsimcoal. $AIC = 2*k - 2*L$, where k is the number of free parameters (i. e. the number of parameters in the *.est file of fastsimcoal) and L is the log-likelihood. After calculating the AIC for each model, these can be directly compared and the best model should have the lower AIC. There are several tutorials available, I found one of them here, for example:https://speciationgenomics.github.io/fastsimcoal2/
I suspect that after comparing models by the AIC the model with no migration will be favored in relation to others, making the argument in the paper stronger.

Thank you for this comment and for sending us the tutorial which was co-written by one of the co-authors of this manuscript (Ravinet). We have now re-ran the analysis including a likelihood and an AIC. The AIC assessment is included as Supplementary Figure 8, and the likelihood assessment as Figure 6. The new analysis led to slightly different results:
From the four most supported scenarios in the novel analysis, the "no gene flow scenario" displays very recent times of coalescence, which may indicate support for ancient admixture (when running scenarios with no gene flow in an empirical data set with evidence of admixture, times of coalescence times become super recent). The three remaining scenarios indicate ancient admixture. We have re-written a paragraph of the discussion, to allude to the possibility of ancient admixture, but that this is not discernible from incomplete lineage sorting. In essence, we did not alter the conclusions, since we argue that the amount of data (~4,000 SNPs) does not allow us to discern clearly between both patterns.

**Paragraph on the discussion** "We find clear evidence for shared genetic variation in Stygocapitella. The most conspicuous evidence for this comes from the admixture analysis, which clearly demonstrates admixed populations in the three species (Figure 5). This evidence is further supported by individuals with intermediate positions in the MDS – a test which is robust to missing data (Figure 4). However, several evidences do not support a preponderant role of recent admixture. First, we obtained no evidence for admixture when using F-statistics, since we find only positive F-values (Table 1). Second, contrary to the expectation of ongoing gene flow, we do not observe higher levels of heterozygosity in sympatric populations (Lubec in the USA, Musselburgh in Scotland, Hausstrand in Germany; Table 3) where individuals of different species are found in the same sediment sample in close proximity (volume ranging from 50-500 cm3). Third, admixture often generates

incongruence between mitochondrial and nuclear markers (Melo-Ferreira et al. 2012; Sloan et al. 2017), which is not seen in single-marker trees (Supplementary Figures 2-5). Fourth, models with exclusive recent admixture are generally poorly supported by the demographic analysis (Figure 6). In contrast, three out of the four most supported demographic scenarios suggest ancient admixture, and one supported no gene flow at all (Figure 6). The scenario with no gene flow inferred coalescent times of 451 and 13,834 generations or years (1 generation is expected to be 1 year, Günter Purschke pers. comm)] which are not compatible with estimates of the splitting age of the three Stygocapitella species (~5-30 million years ago; Cerca et al. 2020b). Given that reduced times of coalescence are a typical signature of simulations that do not account for gene flow, when it has occurred in empirical data (Leaché et al. 2019), it is likely that incomplete lineage sorting alone cannot explain the patterns of shared variation among Stygocapitella species. In other words, the demographic analysis supports a scenario that includes ancient admixture. The three scenarios with ancient admixture vary in the presence or absence of admixture after the second coalescence event (S. subterranea and S. josemariobrancoi): in one scenario, admixture is exclusive to the ancestral branch; in the remaining two, gene flow between S. josemariobrancoi and either S. westheidei or S. subterranea occur. Given the lack of support for on-going gene flow between species by the FST, summary statistics, and F-statistics (Tables 1-3, Supplementary Figures 1-5), admixture may have occurred immediately before or after the speciation event of S. westheidei or S. subterranea, but not in recent times (i.e. the last generations). Furthermore, the occurrence of ancient admixture can affect the inference of recent admixture when not take the phylogeny into account (Malinsky et al. 2018; Ferreira et al. 2020), this may explain the incongruence between some of our analysis. Therefore, while the demographic analysis suggests the occurrence of admixture among S. josemariobrancoi and the other species, future studies are necessary to confidently dissect and determine the role of recent gene flow in the system with independent analyses. For example, these studies will benefit from using whole-genome data to determine whether interspecific divergence in regions of the genome show gene-species tree discordance, thereby dissecting ILS and recent hybridization (Joly et al. 2009; Giska et al. 2019). Also, the demographic analysis favouring a preponderant role of ancient admixture does not exclude the occurrence of ILS, and the beforementioned approach would also allow to clarify the relative contribution of ILS and gene flow to shared patterns of variation among species. In sum, to the extent that we can speculate, our data suggests that shared genetic variance is more likely explained by an evolutionary history including incomplete lineage sorting and ancient geneflow."

Below we copy figure 6 and supplementary figure 8:

[Figure]

**Figure 6:** Demographic scenarios considered. The likelihood of different demographic scenarios is displayed on the Y axis. Based on the estimated phylogeny (Figure 2), we modelled scenarios for (from left to right): 1) geographic gene flow (gene flowbetween S. josemariobrancoi and the ancient lineage, and S. josemariobrancoi and S. subterranea); 2) ancient gene flow (gene flow between S. josemariobrancoi and the lineage before the S. subterranea and S. westheidei split); 3) no gene flow at all; 4) ancient gene flow and gene flow between S. subterranea and S. westheidei; 5) gene flow only between S. westheidei and S. subterranea; 6) gene flow in every possible branch; 7) gene flow in sympatric, European linages; 8) gene flow between S. josemariobrancoi and S. westheidei; 9) gene flow between currently existing lineages; 10) ancient gene flow with gene flow between S. josemariobrancoi and S. westheidei.

[Figure]

**Supplementary Figure 8.** AIC-evaluation of the demographic scenarios. Different models (see main text for details) are depicted in the X axis and have different colours, AIC values are given in the Y axis. Species names are reduced with 's' representing *Stygocapitella subterranea*, 'i' representing *Stygocapitella josemariobrancoi*, and 'w' representing *Stygocapitella westheidei*.

**MM 268-275**: "Finally, we evaluated various demographic scenarios using fastsimcoal2, using the same dataset for the previous analysis which included running fastsimcoal2 (Excoffier et al. 2013). Fastsimcoal2 uses the site-frequency spectrum (SFS) and a coalescent-simulation framework based on an arbitrary user-defined scenario to infer population sizes, strength of gene flow and times of coalescence. To assess these models we calculated AIC and likelihood. Likelihood is calculated by running the 'best parameters' for each specified scenario multiple times and obtaining the distribution of likelihood estimates. AIC was calculated using a script available in https://speciationgenomics.github.io/fastsimcoal2/. "

**Results 420-423:** "Finally, while the AIC assessment provide slightly different results (Supplementary Figure 8), the second and third most supported scenarios are the geographic gene flow and ancient gene flow; being thus in agreement with the likelihood results."

5 - line 280, "module as asymmetric" should be "modelled as asymmetric".

6 - There are weird characters following numbers in the paragraph starting in line 294

All changed accordingly.

On Behalf of all co-authors

José Cerca